# Orbis: Overcoming Challenges of Long-Horizon Prediction in Driving World Models

**Arian Mousakhan**[*]                    **Sudhanshu Mittal**[*]

**Silvio Galesso**[*]          **Karim Farid**[*]          **Thomas Brox**

University of Freiburg, Germany
{mousakha,mittal,galessos,faridk,brox}@cs.uni-freiburg.de

## Abstract

Existing world models for autonomous driving struggle with long-horizon generation and generalization to challenging scenarios. In this work, we develop a model using simple design choices, and without additional supervision or sensors, such as maps, depth, or multiple cameras. We show that our model yields state-of-the-art performance, despite having only 469M parameters and being trained on 280h of video data. It particularly stands out in difficult scenarios like turning maneuvers and urban traffic. We test whether discrete token models possibly have advantages over continuous models based on flow matching. To this end, we set up a hybrid tokenizer that is compatible with both approaches and allows for a side-by-side comparison. Our study concludes in favor of the continuous autoregressive model, which is less brittle on individual design choices and more powerful than the model built on discrete tokens. Project page with code, model checkpoints and visualization can be found here: https://lmb-freiburg.github.io/orbis.github.io

## 1 Introduction

Intelligent agents operate in complex environments by simulating plausible future states based on past observations. This capacity for imagination allows them to plan toward long-term goals [22, 4]. Humans naturally acquire this ability through passive observation and minimal interactions, enabling them to adapt quickly to new and unseen scenarios. Emphasizing the passive observation component of such world models has become particularly popular for the driving world, since large amounts of data exists for this domain. It is attractive to circumvent the manual setup of many perception components by learning the visual representation for decision making via the predictive loss of a world model.

Recent driving world models [17, 1, 25] built on video diffusion models [6] have made major strides towards generating detailed content in high definition and at high frame-rates. However, Fig. 2 highlights that these models only work well for few frames, especially in case of maneuvers that require generating new content, such as turning. Our evaluation, conducted on the generated videos as well as on the estimated ego-trajectories, reveals substantial limitations in how all public driving world models capture state transitions – the key feature of a world model – even though the quality of the generated videos is excellent at first.

While realistic video generation can be a valuable product, it is not the primary objective of world models. As demonstrated in a number of influential works (e.g. World Models [22], Dreamer [23], Cosmos [1], VJEPA-2 [2]), their main purpose is representation learning, planning, and policy learning. The quality of state representations and the accuracy of next-state predictions are therefore paramount, with decoded videos serving primarily as an indicator, e.g. of whether the model can

---

[*]Main Contributors.

39th Conference on Neural Information Processing Systems (NeurIPS 2025).

execute a turn stably. Long-horizon prediction serves as a more meaningful measure of how well the model captures state transitions, while generalization to complex scenarios reflects its ability to model diverse real-world dynamics. Existing models such as Vista and GEM, though conceived with planning as possible application, often fail in non-trivial yet routine situations. In this work we prioritize building a world model with robust state representation and dynamics that handle such cases effectively.

Consequently, a relevant question is whether world models should rely on continuous-space latents or use discrete tokens (similar to LLMs)[15, 10, 86] for representing world states. The current trend for visual generation goes towards diffusion-based (continuous) models [61, 6, 19]. On the other hand, driving world models based on discrete representations and LLM-like objectives seem to have the edge in terms of rollout duration [30, 31]. Among these, the proprietary GAIA-1 model showed no issues with turns and long drives. This observation prompts the question of whether the discrete paradigm is really superior to the other for long-term generation, and whether the continuous space is the reason for the observed shortcomings in the current state of the art.

To address these questions, we introduce a hybrid discrete-continuous tokenizer that is compatible with both types of modeling approaches to be able to compare the two strategies on the same ground. For the quantized-token model we developed a frame-wise autoregressive model based on MaskGIT [10], whereas for the continuous-token model we developed an autoregressive model based on flow matching [46, 49]. Both models were trained from scratch. We also put effort into optimizing the details of the tokenizer. Indeed we find that many of these details are important for the performance of the model acting on quantized tokens. Surprisingly, these details are of little relevance for the continuous modeling approach. Both our models can handle long roll-outs, but the continuous approach yields significantly better results and sets the state of the art by a large margin; see Fig. 1.

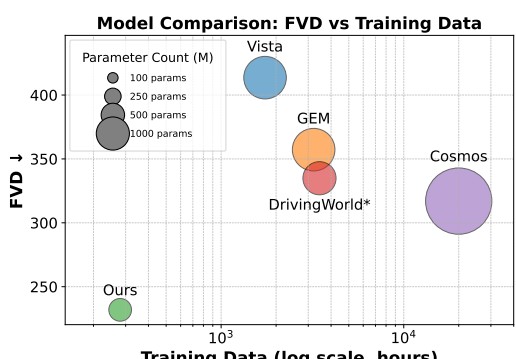

Figure 1: Comparison of model scale, training data volume, and FVD performance of various approaches on the NuPlan-turns dataset. *DrivingWorld is trained on the test dataset nuPlan.

Unlike many prior approaches, our world model is trained using only raw video data without using any extra low-level regularization objectives, such as structural consistency or pseudo-depth supervision. All implicit perception is learned directly from the presented videos. This makes the approach more scalable and establishes a strong foundation for the development of more controllable models.

We also demonstrate that our model can be modified easily to allow ego-motion control via adaptive layer normalization [58]. To this end, we evaluate the trajectories produced by our world model also in ego-motion-control-conditioned settings, where we propose a set of metrics to evaluate realism and coverage of the requested trajectories.

To summarize, (1) we highlight shortcomings of contemporary driving world models and propose additional benchmarking metrics to make these shortcomings more explicit. (2) We propose a hybrid discrete-continuous tokenizer that is compatible with both discrete and continuous prediction losses and allows us to compare both modeling approaches side-by-side. (3) On its basis, we compare continuous and discrete prediction losses on a fair common ground and find a clear advantage in favor of continuous modeling. (4) As demonstrated in Figure 1, the resulting model is much more economical in terms of training data and model size than existing world models. Using only 280 hours of front camera video data, our 469M parameter flow-matching model Orbis already produces state-of-the-art performance on long-horizon rollouts with realistic and diverse trajectories. It excels particularly in challenging driving scenarios.

## 2 Related Work

**World Models.** The ability of world models to do real-world simulation can be useful for policy learning [59, 26], sample efficient RL [23, 24, 79, 42], and representation learning [89]. Previous world models have been limited to gaming [24, 23, 21] and other simulated environments [12]. Recent breakthroughs in video generative modeling [83, 6] have led to future video prediction models - an essential building block for world models.

Multiple driving world models [75, 77, 44] use BEV (Bird's-Eye-View) annotations like depth maps, 3D bounding boxes, road maps to generate new scenarios. DriveDreamer [75] incorporated multi-modal input, such as traffic conditions, text prompts, and driving actions, for future frames and action generation. Many other works [77, 90, 76] extended this idea to multi-view video generation. Some recent works [90, 32] also use LLMs and VLMs [47] for spatial reasoning. Although these models show high quality generation, they rely on heavy external knowledge. Such heavy reliance limits the model's ability to generalize to new environments. In this work, we train a generalizable world model using unannotated front-camera videos and only fine-tune for ego-motion control.

Recent driving world models [36, 82, 17, 1, 25, 30, 66] trained predominantly on raw driving video data have shown the ability to simulate realistic future scenes in unseen environments. DriveGAN [36], among the first works to train on real-world driving data, showed realistic future generation with ego-motion and environment controllability. GAIA-1 [30] further enhanced the quality of future prediction and added controllability through text, in addition to action input. Diffusion-based world models [17, 25] fine-tuned general-purpose pre-trained video generation models like SVD [6] to produce future video predictions at high resolution and high frame rate. Driving world models - Vista [17] and GEM [25] demonstrate high-quality rollouts up to 15 seconds. DrivingWorld [31] further enables longer and more coherent rollouts.

Generative models based on vector-quantized tokens like autoregressive [81, 38] and masked generative models [87, 48, 20], have also demonstrated strong performance in video generation due to their strong capability in modeling dynamics and representation learning. For world modeling, Genie [7] and GAIA-1 [30] have demonstrated generalized world modeling capabilities with interactive control and long-horizon rollouts respectively. We also observe that quantized driving world model can perform long-horizon generation.

**Latent Representation Learning.** VAEs [60] and VQVAEs [73, 15] are foundational autoencoding techniques for learning latent representations used in training latent world models. VAEs produce the continuous latents, commonly used in diffusion [91, 6] and flow matching [66, 61, 33] generative models. VQ-VAE produces discrete (quantized) latent codes for LLM-style autoregressive [15, 37] and masked generative modeling [87, 7]. Following VQGAN work [15], ideas such as product quantization [43, 62, 3], residual quantization [41], multi-scale residuals [70, 53], spectral decomposition [16, 45, 1] have been introduced for image and video generation. Some works also propose hybrid tokenizers [69] that unify the tokenizer model for both discrete and continuous generative models.

## 3 Challenges Faced by Contemporary Driving World Models

Prior works perform well in straight-road driving scenarios but show significantly higher failure rates when faced with difficult maneuvers. For example, in turning events, as rollouts extend over longer horizons, the content generated by these works tends to run out of distribution, producing blurred frames. The degradation of visual semantics and details causes the ego-vehicle to stop prematurely, as the model cannot recover from poorly generated new context. This can be seen in Figure 2, where Vista [17] comes to a halt before 10 seconds. Most of these limitations can be reflected using video quality metrics like FVD [71] computed on several time windows, as shown in Section 5.3.

We found that these models also show unrealistic ego-vehicle behaviors, such as lateral sliding or jitter artifacts. This could be due to the strong priors inherited from general-purposed pre-trained video generation models. Such artifacts can be seen in the trajectories produced by these models, as shown in Fig. 2.

These are not well captured by standard metrics like FVD or JEDi, whose reliance on pretrained human-action or general purpose encoders limits sensitivity to driving dynamics and ego-motion.

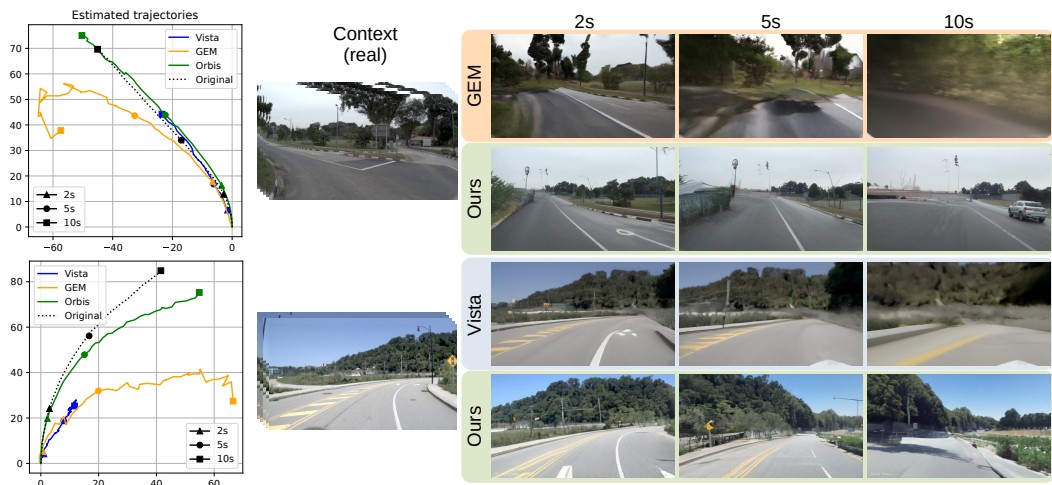

Figure 2: Limitations of state-of-the-art video generation models on turning events. **Left**: The trajectories estimated from the generated videos show that previous approaches either stop prematurely or drift into an unnatural path. **Right**: The quality of the corresponding generated frames degrades over time, as the models struggle to generate the scenery. In contrast, our method tracks the original trajectory curvature and speed more closely, and can generate novel content beyond the unseen horizon. Videos for a larger set of randomly sampled context frames are linked in the Appendix.

This calls for more targeted evaluations. To bridge this gap, we propose a distribution-level trajectory-based evaluation, detailed in Section 5.2.2, that directly quantifies realism and coverage of generated driving behavior compared to a curated dataset of turning events. We evaluate and compare the generated trajectories for Vista, GEM, and our approach, and find the results to confirm our qualitative observations and show the shortcomings of the existing methods.

## 4   Compatible Discrete and Continuous Prediction Models

The above shortcomings all appear in conjunction with approaches based on video diffusion. This modeling approach could be a potential cause of these methods' failure. To enable a fair comparison between discrete and continuous latent world models, we design a hybrid image tokenizer that supports both objectives and allows us to evaluate directly which objective better handles the challenges of long-horizon prediction in a simple and controlled setting. Our study is conducted using two efficient formulations: flow matching for continuous models and masked generative modeling for discrete models.

### 4.1   Hybrid Image Tokenizer

**Preliminary.**   Given an image $\mathbf{I} \in \mathbb{R}^{H \times W \times 3}$, the encoder $\mathcal{E}$ produces a latent $\mathbf{x} = \mathcal{E}(\mathbf{I}) \in \mathbb{R}^{H' \times W' \times d}$, where d is latent channel dimension. The latent $\mathbf{x}$ is then quantized to the closest codebook entry, resulting in $\mathbf{q} = \mathcal{Q}(\mathbf{x}) \in \mathbb{R}^{H' \times W' \times d}$, using a codebook $\mathcal{C} \in \mathbb{R}^{K \times d}$ with $k$ entries. The tokenizer is trained using the VQGAN [15] objective.

**Our design.**   Building upon recent works [43, 85, 88], we design a custom hybrid tokenizer, suitable for both discrete and continuous predictive video modeling. VQ-VAEs [73] typically optimize latent representation learning for pixel-level reconstruction. Prior works show that these representations typically lack desirable properties such as semantic structure [43].To address these limitations, we adopt a factorized token design [3], using separate encoders $\mathcal{E}_s$ and $\mathcal{E}_d$ to produce *semantic* tokens $\mathbf{x}_s$ and *detail* tokens $\mathbf{x}_d$, respectively. The former are obtained via additional distillation from DINOv2 [55] as shown in the Figure 3. Each output is quantized independently using separate codebooks $\mathcal{C}_s$ and $\mathcal{C}_d$, yielding $\mathbf{q}_s = \mathcal{Q}_s(\mathbf{x}_s)$ and $\mathbf{q}_d = \mathcal{Q}_d(\mathbf{x}_d)$. We convert the tokenizer into a *hybrid model* by fine-tuning it with a 50% probability of bypassing the VQ bottleneck during training. This simple modification allows a single tokenizer to support both discrete and continuous latent

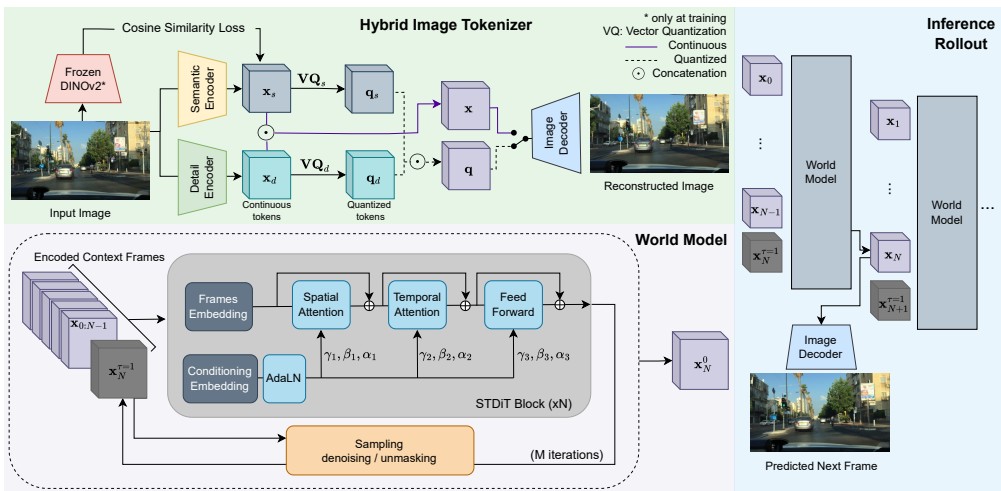

Figure 3: **Image Tokenizer:** The tokenizer provides two semantic and detail representation. These two representations are concatenated and fed into the image decoder and later to the world model. During training the decoder receive continuous or discrete tokens randomly in the fine-tuning phase. **World Model:** To generate the next frame, the model receives either sampled Gaussian noise or fully masked tokens as the target frame, along with encoded context frames. The model progressively denoise or unmask the target frame. This iterative sampling process is repeated to generate target frame. **Inference Rollout:** During inference, the world model autoregressively generates next frame. This process repeats for the desired number of frames in the rollout sequence.

representations. The final continuous representation is $\mathbf{x} = (\mathbf{x}_s; \mathbf{x}_d)$ and the corresponding quantized representation is $\mathbf{q} = (\mathbf{q}_s; \mathbf{q}_d)$. The decoder reconstructs the image from the final representation.

## 4.2 Latent Space World Model

We formulate our world model as a next-frame autoregressive model for both discrete and continuous objectives, as demonstrated in Figure 3. The model receives the context frames $\mathbf{x}_{0:N-1}$ and a target frame $\mathbf{x}_N^{\tau=1}$, initialized by noise or a complete mask. The model predicts the next frame $\mathbf{x}_N^{\tau=0}$ iteratively over multiple $M$ denoising or unmasking steps. During inference rollouts, the model updates its context by appending the most recently generated frame $\mathbf{x}_N^{\tau=0}$, discarding the earliest context frame $\mathbf{x}_0$ of the previous inference step. This sliding-window process is repeated for each next-frame generation to get long-horizon predictions in the latent space (Figure 3). For visualization, each generated latent is decoded into an image using the tokenizer decoder.

In this work, we consider the flow matching [46, 49, 14] objective for the continuous world model and the masked generative modeling objective [10, 87] for the discrete world model.

### 4.2.1 Flow Matching

We follow the flow matching (FM) objective introduced by Lipman [46]: we define a forward trajectory from the data distribution to a standard normal distribution via linear interpolation:

$$\mathbf{x}^\tau = (1 - \tau)\mathbf{x} + \tau\epsilon, \quad \tau \in [0, 1], \quad \epsilon \sim \mathcal{N}(0, I) \tag{1}$$

To use the flow matching objective for next frame prediction, the corrupted target frame $\mathbf{x}_N^\tau$ is conditioned on previous frames $\mathbf{x}_{0:N-1}$. The model predicts $\mathbf{v}(\mathbf{x}_N^\tau; \mathbf{x}_{0:N-1})$, the velocity that would take $\mathbf{x}_N^\tau$ towards the Gaussian prior. We train the model with the following objective as:

$$\mathcal{L} = \mathbb{E}_{\tau\sim[0,1],\, \epsilon\sim\mathcal{N}(0,I)} \left[ ||\mathbf{v}(\mathbf{x}_N^\tau; \mathbf{x}_{0:N-1}) - (\epsilon - \mathbf{x}_N)||^2 \right], \tag{2}$$

At inference time, we sample a noise vector as the new target frame and iteratively transform it towards the data manifold. At each iteration the model calculates the velocity conditioned on context frames and update the target frame as:

$$\mathbf{x}_N^{\tau-\delta_\tau} = \mathbf{x}_N^\tau - \delta_\tau \cdot \mathbf{v}(\mathbf{x}_N^\tau; \mathbf{x}_{0:N-1}) \tag{3}$$

where $\delta_\tau$ is the step size used to update the target frame at time step $\tau$. After integrating from $\tau = 1$ to $\tau = 0$, the resulting latent $\mathbf{x}_N^0$ is the generated next frame in latent space.

### 4.2.2 Masked Generative Model

In the discrete setting, we extend masked generative modeling (MGM), following the MaskGIT objective [10], from image generation to next-frame prediction. The encoded latents of image frames are represented using discrete tokens $\mathbf{q}$.

During training, we apply a binary mask $\mathcal{M} \in \{0,1\}^{H' \times W'}$ to the target frame $\mathbf{q}_N$, resulting in the masked frame $\mathbf{q}_N^{\mathcal{M}} = \mathbf{q}_N \circ \mathcal{M} + [\text{MASK}] \circ (1 - \mathcal{M})$, where $[\text{MASK}]$ is a learned special token. The masking ratio for the whole frame is sampled uniformly from 0% to 100% using a predefined scheduler. The MGM model takes as input the concatenated sequence of context frames and the noised target frame $(\mathbf{q}_{0:N-1}; \mathbf{q}_N^{\mathcal{M}})$, and is trained to predict all the token IDs of the target frame. The training objective is a standard cross-entropy loss, defined as follows:

$$\mathcal{L}_{\text{CE}} = \mathbb{E}_{\mathcal{M}} \left[ - \sum_i \log p_\theta \left( \mathbf{q}_N^{(i)} \mid \mathbf{q}_{0:N-1}, \mathbf{q}_N^{\mathcal{M}} \right) \right], \tag{4}$$

where $i$ indexes over all token positions in the target frame $\mathbf{q}_N$. The model receives discrete token indices from the tokenizer, which discards any pairwise similarity structure among latent tokens. Following [67], to reintroduce this structure, we utilize the similarities between quantized code vectors in the VQ codebooks as an extra regularizer to improve the training objective. At inference, given a fully masked target frame and the context frames $(\mathbf{q}_{0:N-1})$, the model iteratively predicts and replaces masked tokens. We follow, the confidence-based sampling [10] heuristics for unmasking the target frame.

### 4.2.3 Conditioning with Ego-motion

To verify that our model is capable of action control, we implement the option for additional condition signals via adaptive layer normalization [58]. We embed steering angle and speed with a two-layer MLP, and add them to the other condition signals.

## 5 Experiments

### 5.1 Experiment Details

**Datasets.** To train our world model, we use subsets of videos from the BDD100K [84] and OpenDV [82] datasets. As shown in Table 1, we select a limited number of hours from each dataset and extract frames at 10 Hz. In total, we use 280 hours of video data from a combined available total of 2747 hours. For BDD100K, we select the *day-clear* subset of the training set. From OpenDV we exclude night drives via a brightness filter and uneventful ones by the presence of certain words in the original video titles (see Appendix). We then subsample by selecting evenly spaced 30-second clips. To train the tokenizer we additionally select images from Honda HAD [35], Honda HDD [63], ONCE [54], NuScenes [8], and NuPlan [9] to make the dataset diverse. Our dataset primarily consists of daylight scenarios.

Table 1: Overview of the training datasets used for the world model.

| Name | Total (h) | Used (h) | Frames (M) |
|---|---|---|---|
| OpenDV | 1747 | 158 | 5.67 |
| BDD100K | 1000 | 112 | 4.02 |
| Total | 2747 | 280 | 9.69 |

**Tokenizer details.** For the tokenizer, we employ a Transformer-based encoder and a CNN-based decoder. Our tokenizer consists of 234 M parameters and uses two ViT [11] encoders initialized with MAE [27] weights, for the two factorized tokens. To address codebook under-utilization issues, we incorporate L2-normalized codes [85], low-latent dimension [85] and entropy penalty [88]. For further improvement, we fine-tune the model with implicit regularization as proposed in EQ-VAE [39].

**Latent world model details.** Both continuous and discrete models follow a factorized spatial-temporal (ST) transformer architecture [80]. For high-resolution experiments, we replace the spatial block with a Swin [50] transformer block for scalability.

For the FM model, we use DiT [56] with ST transformer blocks (STDiT), where a per-frame causal attention mask is applied to the time-attention layers. To improve generalization and frame generation quality, we drop all context frames 50% of the time. When context frames are present, we augment them with noise 50% of the time, similar to prior work [72, 18, 28]. In order to sample the next frame, we use ODE sampler and take 30 steps. For the MGM model, we also add context noise to improve robustness towards context noise, especially for long-rollouts: we replace 10% of the frames and 10% of the overall tokens with a mask token.

Masked generative models often exhibit flickering artifacts caused by inconsistent predictions across the temporal dimension. We train a lightweight 30M-parameter temporal refinement module to smoothen spatial flickering artifacts. It is U-Net architecture [65], trained using a flow-matching objective. This module operates purely as a post-processing step, on single frames, and does not interfere with the world model. More details are included in the Appendix C.3.

**Training details.** Our higher resolution model operates at $512 \times 288$ and small-scale model at $256 \times 256$. Tokenizer compresses the image spatially by $16 \times$. We train latent models with a context of 5 frames sampled at 5Hz. All small-scale models for ablation studies are trained on only the BDD100K subset for one day on $32 \times$ A100 GPUs. The higher resolution model is trained for 10 epochs over 5 days on $72 \times$ A100 GPUs.

## 5.2 Evaluation

### 5.2.1 Video Generation Quality

We evaluate the quality of the generated videos using FVD [71] and JEDi [52], and we use FID to assess tokenizer reconstruction quality. For comparing with the baselines, we show FVD and JEDi on nuPlan [9] and Waymo [68] datasets with 800 and 400 samples respectively. To evaluate the models in challenging scenarios, we curate a dedicated validation set of turning events (nuPlan-turns), consisting of 400 samples, selected from the nuPlan validation set where the initial yaw rate is at least 0.12 rad/s ($\sim$1 std). FVD results on nuPlan and nuPlan-turns are not comparable, since nuPlan is a much more diverse dataset compared to the specialized nuPlan-turns dataset. All evaluation datasets constitute unseen testing domains for our model and for the baselines, except for DrivingWorld, which contains nuPlan as part of the training dataset. We choose nuPlan over the similar nuScenes due to the latter's irregular sampling rate, which adds an unnecessary confounding factor to the evaluation. Additionally, we evaluate our model using Video Quality Assessment (VQA) metrics including PSNR, SSIM and DOVER[78], see Appendix D.2.

### 5.2.2 Trajectory Quality

To evaluate the realism and coverage of generated videos in a manner well suited to driving scenes, where the realism of ego motion and driving behavior is critical, we propose distribution-level, trajectory-based precision and recall metrics. To this end, we map both real and generated videos to pose sequences using the inverse dynamics model VGGT [74], and evaluate realism and coverage via precision–recall following [40], where the number of nearest neighbors within the distribution determines the distance threshold. To measure distances over the trajectory sets, we use discrete Fréchet distance [13] and Average Displacement Error (ADE) [57], both within the distributions of real and generated trajectories and across

Table 2: Quality of estimated 10s trajectories for Vista, GEM, and our model, evaluated on turning events from nuPlan.

| Model | Frechet | | ADE | |
|---|---|---|---|---|
| | Prec. | Rec. | Prec. | Rec. |
| Vista | 0.39 | 0.45 | 0.25 | 0.48 |
| GEM | 0.33 | 0.54 | 0.27 | 0.47 |
| Ours | **0.47** | **0.56** | **0.41** | **0.51** |

them (see Appendix D.1 for full definitions). The latter is a stricter metric, as the former is agnostic to velocity differences between trajectories.

We compare the quality of generated trajectories for Vista, GEM, and our approach in (Table 2). The results show the limitations of existing models in capturing the underlying distribution of ego motion and driving behavior. All approaches perform worse in terms of ADE, indicating difficulty

Table 3: SOTA results: FVD and JEDi over 6 seconds rollouts@ 5Hz. Numbers of baselines were computed using their official checkpoints. Lower is better. *DrivingWorld (DW) is trained on the test dataset nuPlan and uses ego-motion control as an extra input. Video samples available in Appendix.

| Model | FVD↓ | | | JEDi↓ | | |
|---|---|---|---|---|---|---|
| | nuPlan | Waymo | nuPlan turns | nuPlan | Waymo | nuPlan turns |
| Cosmos [1] | 291.80 | 278.19 | 248.39 | 0.55 | **0.31** | 0.50 |
| Vista [17] | 323.37 | 422.58 | 413.61 | 0.37 | 0.44 | 0.54 |
| GEM [25] | 431.69 | 291.84 | 357.25 | 0.42 | 0.35 | 0.31 |
| DW* [31] | 298.97 | N/A | 334.89 | 1.33 | N/A | 1.50 |
| Orbis (ours) | **134.06** | **167.57** | **239.20** | **0.14** | 0.33 | **0.16** |

in maintaining realistic speeds. Moreover, we achieve the best precision-recall on Fréchet distance, indicating that our predicted trajectories more closely follow the ground-truth paths compared to competing baselines.

## 5.3 Results

**Comparison to SOTA.** We compare our method against the state-of-the-art video world models for autonomous driving: Vista [17], GEM [25], DrivingWorld (DW) [31], and the more general-purpose Cosmos [1] in its autoregressive Predict1-4B version. We focus our comparison on steering-free unconditional generation, i.e. with previous visual observations as sole conditioning, with the exception of DrivingWorld which requires the past trajectory. We use a context size of five frames for Vista and DrivingWorld, one for GEM and nine for Cosmos – as per their respective designs. The input control for DrivingWorld is implemented for nuPlan's data format.

Results are shown in the Table 3, for a prediction horizon of 6 seconds at 5hz. The qualitative results for 20s are shown in Figure 5 (more are included in Appendix). Our method outperforms other driving world models on all three benchmarks and both metrics, except for Cosmos on Waymo according to the JEDi metric. We further compare results for long-horizon video prediction, shown in Figure 4 on nuPlan-turns dataset. For each method, FVD is computed over the entire predicted video in a windowed manner, where each window contains 20 frames sampled at 5hz. Results show that Orbis based on flow matching outperforms all baselines and maintains stable performance over long-horizon prediction of up to 20 seconds. The discrete counterpart Orbis-MG based on MaskGIT, shows suboptimal performance for shorter horizons but scales well over long horizons, surpassing all previous works for the last two

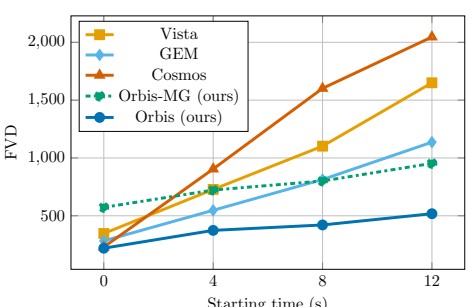

Figure 4: Video quality (FVD) over consecutive 4s time windows on nuPlan-turns. The x axis shows the starting time of the evaluated time window.

windows. As discussed earlier, previous works perform well in short horizons but struggle with long-horizon predictions. GEM has higher FVD scores for short-horizon due to its single-frame conditioning design but performs relatively better on long-horizon predictions (more details in the Appendix E).

**Ego-motion Control and Evaluation** As a proof-of-concept for ego-motion control, we fine-tune a copy of Orbis model for two epochs on 75h of nuPlan videos and IMU data. Following previous literature [17, 25, 31], we evaluate the resulting model by computing the ADE [57] between true and generated trajectories, estimated with VGGT. We compare the ADE of the same model with and without steering on 400 5s long nuPlan validation sequences in Table 5. Better trajectory tracking under ego-motion conditioning indicates some degree of controllability – though in a preliminary setting. Indeed, conditioning capabilities for related models are well documented [56, 4].

Table 4: Tokenizer ablation. rFID is computed on 10k BDD100k images. FVD on 200 sequences of 60 frames.

| DINO | TF | Vocab Size | rFID ↓ | FVD ↓ Orbis-MG | FVD ↓ Orbis |
|------|-----|-----------|--------|----------------|-------------|
| ✗ | ✗ | 4096 | 9.33 | 1331.28 | 240.34 |
| ✓ | ✗ | 4096 | 12.17 | 1214.34 | 248.79 |
| ✓ | ✓ | 2×4096 | 9.10 | 533.28 | 246.11 |

Table 5: Ego-motion control: effect on the average error between real and generated trajectories.

| Model | ADE ↓ |
|-------|-------|
| Unconditional | 5.20 |
| + ego-motion | 2.40 |

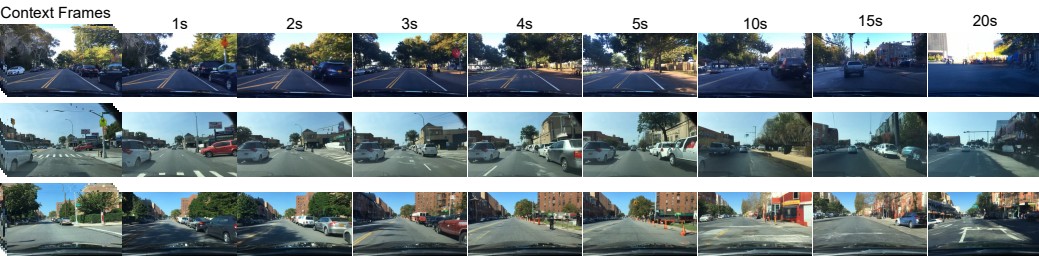

Figure 5: Qualitative results of the Orbis model over 20-sec rollouts (zoom-in for details). Videos and more samples available on the webpage.

**Effect of tokenizer design.** For the discrete model, adding DINO distillation to the image tokenizer, similar to GAIA-1 [30], leads to lower FVD, as shown in Table 4. However, the key factor to enable long-horizon prediction for the discrete model is token factorization. Usage of DINO distillation even leads to a worse rFID (reconstruction FID). However, token factorization annihilates this difference. Interestingly, while the factorized tokenizer with DINO distillation is very important for the discrete model, the continuous model is robust to these design changes, showing no large change in FVD. These experiments were conducted in the small-scale setting.

**Shortcomings of discrete space modeling.** Despite being capable of relatively long rollouts, the videos produced by Orbis-MG on average stop earlier than its flow matching counterpart, and their duration is very sensitive to the sampling heuristics. We investigated this phenomenon and found that at each location, the model's classifier chooses the exact same token as the last context frame approximately 45% of the times (this number is 29% for original encoded frames). This is likely because in the discrete space content copying is an obvious and most rewarding choice. While this phenomenon can be mitigated with regularization like context augmentation and a token-similarity based loss, it does not get fully resolved. Additionally, the discrete model fails to capture small motions of objects which is crucial for driving scenarios - thereby limiting the expressivity of the world model.

**Consistency over architectures and sampling budgets.** To assess the generality of our findings, we evaluate Orbis-FM and MG using three Transformer architectures – DiT [56], STDiT (our default), and CDiT [5], and with different sampling budgets. All models were trained for one day on 32 A100 GPUs in a small-scale setting, and FVD scores were computed on 200 generated BDD validation videos (60 frames at 5 fps). As shown in Table 6, Orbis-FM consistently outperforms Orbis-MG across all architectures and inference settings (12 and 30 steps). Orbis-FM shows greater consistency across architectures than Orbis-MG. The ST architecture shows the best results for both models, with Orbis-FM (ST, 30 steps) achieving the best performance overall.

Table 6: FVD scores for Orbis-MG and Orbis-FM model with different architectures. The FVD is over 60 generated frames of BDD val set.

| Model | Architecture | 12 steps | 30 steps |
|-------|-------------|----------|----------|
| Orbis-MG (discrete) | ST | 533.3 | 571.7 |
| | DiT | 769.0 | 981.3 |
| | CDiT | 1552.3 | 1718.9 |
| Orbis-FM (conti.) | ST | 360.9 | 246.0 |
| | DiT | 345.4 | 274.2 |
| | CDiT | 410.2 | 246.1 |

# 6 Discussion

We investigated an important shortcoming of contemporary driving world models: their struggle with the generation of new content, which makes long roll-outs, turning maneuvers, and realistic trajectories impossible. We introduced an evaluation benchmark and metrics to quantify these problems and tested the hypothesis that modeling in continuous space is the cause of this problem. We found that this is not the case. Based on a side-by-side comparison with a fully compatible hybrid tokenizer, we obtained two driving world models that both provide long roll-outs. However, the continuous model based on flow matching performs much better and sets the new state of the art. The resulting world model has only 469M parameters and was trained on only 280 hours of raw video data. This is significantly less than existing models. At the same time, the approach is perfectly scalable. In contrast to many other recent approaches, it only requires raw video data for training. While we were limited on computing resources for scaling the model ourselves, we expect further improvements when scaling the model parameters, the hours of observed data, the image resolution, and the context length.

**Limitations:** While our investigation showed that world models built in continuous space are advantageous over models built in a quantized token space, we were not able to uncover the reason why the much larger public video diffusion models fail on long roll-outs. One of the possible reason for this could be that these models are typically (but not always) derived from a pretrained Stable Video Diffusion model. This could introduce biases in the representation, which are problematic for learning relevant state transitions and generating long roll-outs for driving case. We will analyze this in more detail as future work.

Apart from this analytic question, our world model has still several limitations, many of which can probably be mitigated by scaling the model along multiple axes. Detailed content, such as traffic lights and street signs, are not yet generated reliably. The traffic actors do not always follow the traffic rules. While our model shows a good diversity when running multiple roll-outs with the same context, the generated trajectories do not seem to represent the true probability distribution. While we showed that conditioning modality can be added flexibly to the model, we did not yet investigate the effectiveness of our model on downstream tasks, such as short-term decision making or planning.

**Societal Impact:** In this work, we contributed to the building of world models – a technology, which may enable more reliable and cost-efficient autonomous driving and may play a key role in interactive robotics. In its present state, the research is still in its infancy and results that will affect society will still require a few years.

## Acknowledgement

This work is funded by the German Federal Ministry for Economic Affairs and Energy within the project "NXT GEN AI METHODS" (19A23014R). The authors gratefully acknowledge the Gauss Centre for Supercomputing e.V. (www.gauss-centre.eu) for funding this project by providing computing time through the John von Neumann Institute for Computing (NIC) on the GCS Supercomputer JUWELS [34] at Jülich Supercomputing Centre (JSC). The project was accomplished under GCS compute grants - *genai-ad* and *nxtaim*. The compute used in this project is also funded by the German Research Foundation (DFG) - 417962828, 539134284. The authors acknowledge support from the state of Baden-Württemberg through bwHPC. We thank Marcel Aach and Sabrina Benassou for their technical support and assistance with the GCS HPC cluster. We thank Rajat Sahay for valuable discussions and feedback during the development of this work.

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

# Appendix

## A  Video Rollouts

We include qualitative examples in video form, embedded in the `https://lmb-freiburg.github.io/orbis.github.io/` page. We have divided the qualitative examples into sections.

### A.1  Comparison with the state-of-the-art

Here we show videos generated by our method beside those generated by the baseline approaches (Vista [17], GEM [25], DrivingWorld [31], Cosmos [1]), for the same initial condition frames. We include videos generated by our method for both continuous (FM) and discrete (MG) version.

These videos showcase the superiority of our model in dealing with content generation after turning events. Orbis (FM) can generate more realistic scenes and objects than its discrete (MG) counterpart. Moreover, in the fifth scene our model is the only one to halt at a stop sign, generating the passing of a pedestrian and a car.

Along the generated videos, we display the estimated trajectories for Orbis, Vista, and GEM. These show the unrealistic ego-motion that the SVD-based methods produce in some cases. Trajectories are estimated using the VGGT model [74].

### A.2  Performance in different scenarios

Here we show our videos generated by our model on straight drives, turns, and urban scenes. Moreover, we show how our model can generate diverse videos when starting from the same initial condition frames.

### A.3  Randomly sampled videos

Here we show randomly sampled videos, all generated by our model, for nuPlan, Waymo, and BDD100K. The first two are out-of-domain, whereas the last is in-domain w.r.t. the model's training data.

Even though our approach can generate videos from out-of-domain condition frames, its rollouts stop more often on nuPlan and Waymo samples, compared to BDD100K.

## B  Dataset Details

### B.1  OpenDV

We filter the training videos from OpenDV by brightness and by video title. We discard all videos containing any of the following words in their original title: `night`, `scenic`, `interstate`, `nature`, `desert`, `park`, `walking`. We then discard all videos with an average pixel value below 90 in a [0, 255] range, in order to keep consistency with the selected BDD100K subset.

From the resulting 1337 videos we then discard the first and last 60 seconds (to avoid text and other overlays) and extract a total of 19398 30-second long clips.

### B.2  Validation Sets

Here we describe how we obtained the validation sets used in the paper. We will release the annotation files needed to reproduce the validation sets.

#### B.2.1  nuPlan

For this benchmark, we use the validation set of nuPlan [9], at its original sampling rate of 10Hz. We select the validation samples by ensuring a distance of 8 seconds between their starting frames and a length of at least 20 seconds worth of real frames available for evaluation.

The total resulting samples are 5878. Due to the computational cost of generating videos for all approaches we evaluate on the first 800 samples.

### B.2.2 nuPlan-turns

For this benchmark, we use the validation set of nuPlan [9]. We select the starting frames of the validation samples based on three criteria:

- a distance of at least 3 seconds between consecutive samples,
- at least 40 seconds worth of real frames available for evaluation,
- an initial yaw rate of at least 0.12 rad/s, equivalent to approximately 1 standard deviation.

We evaluate on 400 of the resulting 416 samples.

### B.2.3 Waymo

This benchmark is based on the validation set of the Waymo Open Datset [68], at its original sampling rate of 10Hz.

We select the validation samples by ensuring a distance of 2 seconds between their starting frames and a length of at least 15 seconds worth of real frames available for evaluation. We use 400 of the resulting 406 samples selected with these criteria.

## C  Model Details

### C.1  Latent world model: Training details

Both continuous and discrete models follow a spatial-temporal Transformer architecture. ST-Transformer blocks [80] have interleaved spatial and temporal attention layers. For high-resolution experiments, we replace the spatial block with a Swin Transformer [50], leveraging windowed attention for efficiency. Our transformer architecture consists of 24 ST-blocks with a hidden dimensionality of 768, split across 16 attention heads. We train models with a context of 5 frames sampled at 5Hz, using the AdamW [51] optimizer with a learning rate of $5 \times 10^{-5}$.

### C.2  Flow matching

We modify the DiT [56] to a STDiT architecture by decomposing temporal and spatial attention. As shown in Table 7, the STDiT not only achieves a better FVD but also the frame quality, measured by FID, degrades more slowly over time.

We compute the standard deviation of the training set's encoded representations and normalize each frame by dividing by this value, ensuring unit variance across inputs  [64]. This normalization occurs for detail and semantic tokens independently. To improve generalization and frame generation quality, we drop context frames 50% of the time. This number reduces to 10% after 5 epochs of training. When context frames are present, we augment them with noise 50% of the time, similar to prior work [72, 18, 28]. In order to sample the next frame, we use ODE sampler and take 30 steps [46].

Table 7: Comparison of DiT and STDiT performance. Metrics are computed over 200 sequences, each consisting of 120 generated frames, using the BDD100K dataset.

| Name | FVD ↓ | FID ↓ frame 30 | FID ↓ frame 60 | FID ↓ frame 90 | FID ↓ frame 120 |
|---|---|---|---|---|---|
| DiT | 287.03 | 81.46 | 91.06 | 98.45 | 101.91 |
| STDiT | 273.69 | 77.98 | 85.53 | 89.99 | 89.80 |

### C.3  Masked generative modeling

Here, we explain the extra regularizer which is added to improve the training process of the discrete model.

Table 8: Overview of the objectives used in three training phases of the hybrid image tokenizer. $\lambda_D = 2.0, \lambda_{EQ} = 0.25, \lambda_G = 0.1$.

| | Total | Trainable | Train Mode | Objectives | #Epochs |
|---|---|---|---|---|---|
| . | Phase-1 | Full model | discrete-only | $L_{rec} + L_{VQ} + L_{per} + \lambda_D L_{DINO}$ | 12 |
| | Phase-2 | Full model | discrete + cont. | Phase-1 + $\lambda_{EQ} L_{EQ}$ | 3 |
| | Phase-3 | Decoder-only | discrete + cont. | $L_{rec} + L_{per} + \lambda_G L_{GAN}$ | 5 |

Since, the model takes discrete token indices as input from the tokenizer, it discards any pairwise similarity structure of the latent tokens. To reintroduce this structure, we utilize the similarity matrix $\mathbf{S} \in \mathbb{R}^{K \times K}$ over the $K$ codebook vectors and let $s_i = \mathbf{S}_i$ be the $i^{th}$ row corresponding to the ground-truth token index $i$. Formally, letting $u_i \in \mathbb{R}^K$ be the model's output logits for target token with index $i$, we define

$$p_i^o = \frac{e^{u_i/T}}{\sum_j e^{u_j/T}}, \quad p_i^t = \frac{e^{s_i/T'}}{\sum_j e^{s_j/T'}} \tag{5}$$

where $T$ and $T'$ are temperature hyperparameters and $p_i^t$ is treated as soft-target for the model output. The objective is to minimize the KL-divergence between $p_i^o$ and $p_i^t$ as

$$\mathcal{L}_{\mathrm{KD}} = T \, T' \sum_{i=1}^{H'W'} D_{\mathrm{KL}}\left(p_i^t \, \| \, p_i^o\right). \tag{6}$$

This is similar to knowledge distillation objective [29], which aims to enrich relational information by using soft-targets instead of hard one-hot labels and are known to improve data efficiency and generalization. The overall model training objective is $\mathcal{L} = \mathcal{L}_{CE} + \lambda \mathcal{L}_{KL}$.

We use $T = 2, T' = 0.2$ and $\lambda = 0.5$ for our experiments.

**Refinement module.** The discrete masked generative model struggles to maintain temporal coherence across the full spatial extent of each frame. While it captures important temporal connections to keep the motion of objects consistent across frames and often predicts token with correct semantic property, it predicts tokens with inconsistent appearance. This is likely due to the limitations of heuristic-based unmasking scheme. These inconsistencies result in flickering artifacts, which degrades the quality of the video. These artifacts negatively impacts FVD performance, especially for long-horizon prediction, where the corrupted predictions are reused as context. To remedy these artifacts and compare FVD fairly with continuous baselines, we introduce a small video refinement model comprising of 30 M parameters. This refinement module is only a post-processing unit and does not affect the world model learning. It follows a U-Net architecture with 12 3D-convolutional layers and operates in the latent space. It takes four predicted frame latents from the world model as input and outputs the refined continuous latents.

It is trained directly on the tokenizer encoder output, where it predicts clean continuous latents from the corrupted quantized tokens from the tokenizer. To simulate noise, 20% of the quantized tokens are replaced with randomly picked top-1000 tokens based on the similarity matrix. Ground-truth continuous latents from the hybrid image tokenizer serve as training targets. The model is trained with a flow-matching objective to denoise corrupted latents. At inference, refinement is applied in a sliding-window manner over 4 frame latents, sliding one frame at a time. Only the last predicted frame latent is retained and updated. We use ODE sampler and take only 1 step. The refined latents are decoded by the tokenizer to produce the final image frame.

### C.4 Tokenizer training details

We initialize the VIT-Base encoder with pretrained MAE weights. Both encoder branches combined consists of 171.6 M parameters. The CNN-based decoder architecture is based on VQGAN [15] tokenizer and consists of 44.8 M parameters. We use 16-dim latents, each for semantic and detail codebooks. For the final model, we train the quantized version of the image tokenizer with codebook

size of 16384 for each codebook. The model training has three phases. First phase is similar to VQGAN training, but without the adversarial loss [1]. In the second phase, we fine-tune with scale-equivariance regularization [39]. We only fine-tune the decoder in the third phase with the adversarial loss. Three phases in total comprise of 20 epochs of training. Phase-2 and Phase-3 are trained with a mix of discrete and continuous latents (includes VQ for discrete) to enable corresponding types of world modeling, as shown in Fig. 3 of the main manuscript. In the mixed fine-tuning phases, 50% mini-batches are trained with discrete latents and 50% with continuous latents. Hyperparameters and objective details of three phases are included in Table 8. $L_{rec}$ refers to $L_1$ reconstruction loss, $L_{per}$ refers to perceptual loss, $L_{EQ}$ refers to scale-equivariance regularization loss, $L_{GAN}$ refers to the adversarial loss and $L_{VQ}$ refers to the vector-quantization objectives including codebook and commitment losses.

The model is trained with a mix of 7 datasets comprising of 2.49 M images. OpenDV dataset accounts for around 90% of the dataset. The split across all datasets included for tokenizer training is shown in Table. 9. We select only daylight images for the dataset.

Table 9: Tokenizer dataset overview.

| Name | Frames |
|---|---|
| OpenDV [82] | 2.26 M |
| BDD100K [84] | 158.6 K |
| Honda HAD [35] | 5.1 K |
| ONCE [54] | 14 K |
| Honda HDD [63] | 5 K |
| NuScenes [8] | 3 K |
| NuPlan [9] | 47.4 K |
| Total | 2.49 M |

## D  Evaluation Metrics

### D.1  Trajectory evaluation metrics

To evaluate the distributional fidelity of generated trajectories, we use two primary metrics: *pointwise error* and *curve similarity*. These metrics serve as distance measures to evaluate *distributional fidelity and coverage* using precision–recall [40] in the driving trajectory space relevant to world model evaluation. Specifically, we represent a driving trajectory as a sequence of extrinsic transformation matrices $\mathcal{T} = (\mathbf{T}_1, \ldots, \mathbf{T}_T)$, where each $\mathbf{T}_t$ comprises a rotation (orientation) $\mathbf{R}_t \in SO(3)$ and a translation (position) $\mathbf{p}_t \in \mathbb{R}^3$, arranged as $\mathbf{T}_t = [\mathbf{R}_t, \mathbf{p}_t; \mathbf{0}, 1]$. For computing the ADE and Fréchet distances, we consider only the planar positions $\mathbf{p}_t \in \mathbb{R}^2$. Other parameters within $\mathbf{T}_t$, such as the rotation $\mathbf{R}_t$, can additionally be utilized to assess realism aspects like turning behavior and orientation evolution over time.

**Average Displacement Error (ADE).** Given a predicted trajectory $\hat{\tau} = (\hat{\mathbf{p}}_1, \ldots, \hat{\mathbf{p}}_T))$ and a ground-truth trajectory $\tau = (\mathbf{p}_1, \ldots, \mathbf{p}_T)$, with positions $\hat{\mathbf{p}}_t, \mathbf{p}_t \in \mathbb{R}^2$, the ADE is defined as the average Euclidean distance between corresponding points:

$$\text{ADE}(\hat{\tau}, \tau) = \frac{1}{T} \sum_{t=1}^{T} \|\hat{\mathbf{p}}_t - \mathbf{p}_t\|_2.$$

This metric quantifies pointwise deviations and is sensitive to minor spatial misalignments.

**Discrete Fréchet Distance.** The discrete Fréchet distance assesses the alignment cost between two trajectories while preserving their temporal ordering:

$$\text{FD}(\hat{\tau}, \tau) = \min_{\alpha, \beta} \max_{i=1,\ldots,m} \|\hat{\mathbf{p}}_{\alpha(i)} - \mathbf{p}_{\beta(i)}\|_2,$$

where $\alpha, \beta$ are non-decreasing mappings from trajectory indices to points. This metric emphasizes structural similarity and penalizes shape mismatches more robustly than ADE.

**Precision and Recall in Trajectory Embedding Space.** To evaluate the distributional alignment between real and generated trajectories, we utilize the precision–recall framework [40]. We first map videos into the trajectory space using an inverse dynamics model (VGGT [74]). Let $\mathcal{R} = \{\mathbf{r}_i\}_{i=1}^N$ and $\mathcal{G} = \{\mathbf{g}_j\}_{j=1}^M$ denote the planar positions trajectories of real and generated trajectories, respectively. For each real trajectory $\mathbf{r}_i$, define the threshold $\delta_i^\mathcal{R}$ as the distance to its $k$-th nearest neighbor in the real trajectory space $\mathcal{R} \setminus \{\mathbf{r}_i\}$. Conversely, for each generated trajectory $\mathbf{g}_i$, define $\delta_j^\mathcal{G}$ as the distance to its $k$-th nearest neighbor in the generated trajectory space $\mathcal{G} \setminus \{\mathbf{g}_j\}$. Precision and recall for a distance metric $d(\cdot, \cdot)$ (e.g., Fréchet) are then defined as:

$$\text{Precision} = \frac{1}{M} \sum_{j=1}^M \mathbf{1} \left[ \exists \mathbf{r}_i \in \mathcal{R} \text{ s.t. } d(\mathbf{g}_j, \mathbf{r}_i) < \delta_i^\mathcal{R} \right], \tag{7}$$

$$\text{Recall} = \frac{1}{N} \sum_{i=1}^N \mathbf{1} \left[ \exists \mathbf{g}_j \in \mathcal{G} \text{ s.t. } d(\mathbf{r}_i, \mathbf{g}_j) < \delta_j^\mathcal{G} \right], \tag{8}$$

This adaptive, density-aware thresholding enables reliable evaluation of both fidelity (precision) and coverage (recall), offering a realistic reflection of how well the generated trajectories capture the diversity and structure of real-world driving behavior.

## D.2 VQA evaluation metrics.

**PSNR and SSIM.** As shown in the Table 10, we computed average PSNR and SSIM metrics over two shifted video windows of 10 frames (i.e. 2 seconds at 5fps) over 400 generated videos on nuPlan-turns evaluation benchmark. We resize generated videos of all methods to same resolution for a fair comparison. We observe that Orbis model performs marginally better than other methods over the first window. However, all methods converge to similarly low numbers in the second window.

Table 10: VQA metrics: PSNR and SSIM on windowed video of 10 frames over 400 generated videos on nuPlan-turns.

| Model | PSNR frames 0-9 | PSNR frames 10-19 | SSIM frames 0-9 | SSIM frames 10-19 |
|---|---|---|---|---|
| Cosmos | 17.29 | 13.38 | 0.47 | 0.38 |
| GEM | 14.85 | 13.73 | 0.42 | 0.41 |
| Vista | 15.04 | 13.70 | 0.44 | 0.42 |
| DW | 17.67 | 14.70 | 0.44 | 0.38 |
| Orbis | 18.72 | 14.75 | 0.52 | 0.42 |

**DOVER.** We evaluated the models on the DOVER [78] score, which is also used in the data curation/filtering pipeline of Cosmos. In the Table 11, we report a comparison of the DOVER scores on the nuPlan Turns dataset, computed over 17s long videos. We include both the results for the full Cosmos pipeline (including the extra 7B-parameters diffusion refiner, "Cosmos+ref"), and for the pure Cosmos world model with a non-generative decoder.

Table 11: Blind VQA DOVER metric on 400 generated videos of 17s on nuPlan-turns.

| Method | DOVER↑ |
|---|---|
| Cosmos | 19.94 |
| Cosmos+refinement | 28.99 |
| GEM | 19.76 |
| Vista | 21.14 |
| DW | 21.92 |
| Orbis | 21.34 |

### D.3 FVD evaluation.

We compute FVD is three formats to evaluate both short and long-horizon predictions. We compute short-horizon prediction over the first 6 seconds of predicted video. Results of short-horizon are shown in Table 2 in the main manuscript and Table 12 in the Appendix. Long-horizon FVD is evaluated in two ways: cumulative and chunked. In cumulative FVD evaluation, FVD is computed on increasing video lengths starting from 4 seconds, up to 16 seconds. Results for cumulative-FVD on nuPlan-turns dataset are shown in Fig. 6a in the Appendix. Chunked-FVD is computed on consecutive 4 seconds windows taken at different starting timestamps, shown in Fig. 4 in the main manuscript and Fig. 6b in the Appendix on nuPlan-turns and nuPlan evaluation sets respectively.

## E More results

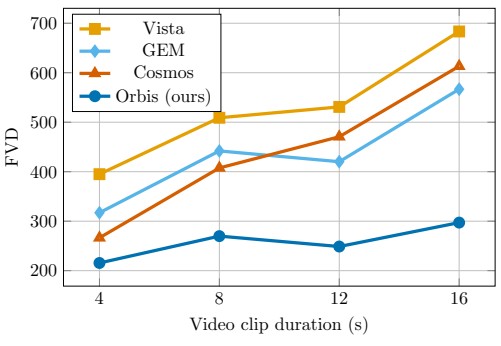 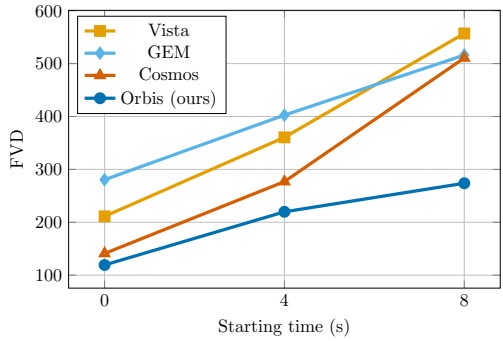

(a) Video quality (FVD) over accumulated 4s time windows on nuPlan-turns. The x axis shows the video clip duration in seconds.

(b) Video quality (FVD) over consecutive 4s time windows on standard nuPlan. The x axis shows the starting time of the evaluated time window.

Figure 6: (a) Cumulative FVD on nuPlan turns on 400 samples and (b) Chunked FVD on nuPlan standard evaluation set on 800 samples.

Table 12: FVD over 6 seconds at original frame rate of different baseline methods. Lower FVD is better. *DrivingWorld (DW) is trained on the test dataset nuPlan and uses ego-motion control as an extra input.

| Model | fps | nuPlan | Waymo | nuPlan turns |
|---|---|---|---|---|
| Cosmos [1] | 10 | 210.56 | 249.08 | 244.80 |
| Vista [17] | 10 | 289.95 | 351.42 | 353.27 |
| GEM [25] | 10 | 348.36 | 218.61 | 318.73 |
| DW* [31] | 5 | 298.97 | N/A | 334.89 |
| Orbis (ours) | 5 | **132.25** | **180.54** | **231.88** |

**FVD at original frame rate.** Originally, the previously published models were trained and evaluated with different frame rates. The main manuscript evaluated all models at 5hz for a fair comparison, skipping alternative frames if the prediction frame rate is 10hz. Here, we also include FVD scores at original prediction frame rates over 6 seconds rollouts, shown in Table 12. The models evaluated at 10hz achieve lower FVD scores than their 5hz counterparts. Despite FVD's sensitivity to frame rate, our model at 5hz still outperforms prior approaches evaluated at higher frame rates.

**Cumulative FVD on nuPlan-turns.** Figure 6a shows results for cumulative-FVD scores on the nuPlan-turns evaluation set. Our proposed model consistently outperforms other baselines, showing a strong scalable behavior as the prediction window extends.

**Chunked FVD on nuPlan.** We also evaluate chunked FVD on nuPlan evaluation set using 800 samples, shown in Fig. 6b. Our model consistently outperforms all baseline across all video windows. Cosmos performs relatively well on early prediction windows but degrades very quickly over later windows. In contrast, GEM performs worse in early windows, but extends well for later windows. We observe GEM suffers in the early prediction windows likely due to its single frame context, which causes it to deviate from the ground truth trajectory earlier than other baselines. However, GEM generates better content in later windows, outperforming other baselines over extended predictions.

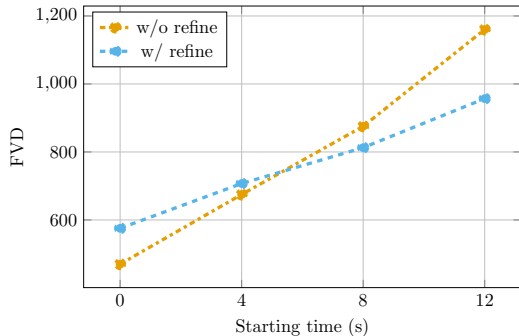

Figure 7: Effect of refinement module in masked generative modeling (orbis-MG model). Video quality (FVD) over consecutive 4s time windows on nuPlan-turns. The x axis shows the starting time of the evaluated time window.

**Effect of refinement module.** The refinement module is design to reduce flickering artifacts caused by imprecise decoding of frames in masked generated modeling. We find that refinement module is effective for long-horizon predictions, where the context is usually corrupted. However, the module has a detrimental effect on short-horizon performance. Fig. 7 shows FVD on nuPlan-turns in a windowed (chunked) evaluation, with and without the usage of refinement module. We observe that the refinement module shows improvement for long-horizon prediction, especially longer than 6 seconds.

**Inference speed and memory requirements.** Table 13 compares the average times (fps) needed to generate a frame, and the required GPU memory. Orbis has the best throughput compared to all other methods. This advantage can be attributed to the smaller size of the Orbis model. By parallelizing Orbis' computation in batches we can achieve an higher throughput. GEM and Vista are based on the same architecture but use different sampling protocols trading off FPS and VRAM.

We also compare the inference speed of our discrete Orbis-MG model and the continuous Orbis-FM model. We also report the GPU (VRAM) memory requirements for both methods during inference. Orbis-MG shows better inference speed compared to the Orbis-FM model. However, since Orbis-FM achieves significantly better video generation performance, it remains the default choice despite the speed advantage of the discrete model.

Table 13: Comparison of inference speed and VRAM memory requirements of different models.

| Method | FPS↑ | VRAM (GB)↓ |
|---|---|---|
| Cosmos | 0.18 | 29 |
| Vista | 0.58 | 86 |
| GEM | 0.44 | 45 |
| DW | 0.25 | 10 |
| Orbis-FM (ours) | 0.70 | 24 |
| Orbis-MG (ours) | 0.85 | 21 |

