# OpenReview forum: "Overcoming Challenges of Long-Horizon Prediction in Driving World Models"
_NeurIPS.cc/2025/Conference — NeurIPS 2025 poster_

### Official Review · Reviewer_NJY5 · 2025-06-29

**Clarity:** 3
**Significance:** 3
**Originality:** 2
**Rating:** 4
**Confidence:** 4

**Summary:**

The paper propose a fair comparison between two modeling paradigms for world models: discrete token-based models and continuous models. To enable this comparison, the authors introduce a novel hybrid tokenizer that can be used with both approaches, ensuring that differences in performance are due to the modeling strategy rather than data representation.

The experiments show that while both models can handle long rollouts, the continuous model is significantly more robust and achieves state-of-the-art performance, especially in challenging scenarios. Besides, the paper also introduces new benchmarking metrics to better capture the shortcomings of existing models.

**Questions:**

- More ablation study focusing on the architecture would be useful to fully understand the effectiveness of continuous tokens.
- Additional metrics, such as inference speed, would be helpful for understanding the effectiveness of continuous tokens.

**Ethical Concerns:**

["NO or VERY MINOR ethics concerns only"]

**Final Justification:**

While the rebuttal have addressed most of my concerns, some concerns remain — particularly that the core modeling approaches are adaptations of existing methods with limited novelty. Given this, I am maintaining my original score.

**Limitations:**

yes.

**Paper Formatting Concerns:**

No.

**Quality:**

3

**Strengths And Weaknesses:**

Strengths:
- The paper introduce new metrics that better capture real-world performance, especially in difficult cases like turning and urban traffic.
- The proposed continuous model achieves state-of-the-art performance with less data and a smaller model size than many competitors, demonstrating both efficiency and effectiveness.

Weaknesses:
- Will the different architectures used for FM and MGM affect the comparability of their performance? Additional ablation studies would provide more convincing evidence.
- While the hybrid tokenizer and empirical comparison are valuable, the core modeling approaches (autoregressive, flow-matching, MaskGIT) are adaptations of existing techniques.

---

> ### Author Rebuttal · Authors · 2025-07-30
>
> Thank you for the positive feedback and thoughtful questions. We provide answers to each question below.
>
> > **Q1:** Ablation study on the performance of FM and MGM model with different architectures.
>
> We evaluated three Transformer architecture variants: DiT [Peebles et al. 2022 ICCV 2023], STDiT (our default), and CDiT [NWM Bar et al., CVPR 2025] - with both the FM and MGM models. All models were trained in a small-scale setting for one day on 32 A100 GPUs and we report FVD scores.
> As shown in the table below, Orbis-FM model consistently outperforms Orbis-MG across all three architectures in terms of FVD. We evaluated both methods using 12 and 30 inference denoising (unmasking for MGM) steps.
> The Orbis-FM model also shows relatively stable performance across different architectures, whereas Orbis-MG shows larger variance.
> ST architecture gives the best performance for both model types across different architectures.
> Overall, Orbis-FM with ST architecture achieves the best performance with 30 steps, as used in the paper.
> FVD scores are computed over 200 generated videos on the BDD validation set, each over 60 frame rollouts at 5 fps.
> We will include this ablation table in the revised version of the paper.
> | Model      | Architecture | 12 Steps | 30 Steps |
> |------------|:--------------:|:--------:|:--------:|
> | Orbis-MG   | ST           |  533.3   | 571.7   |
> |            | DiT          |   769.0    |   981.3    |
> |            | CDiT         | 1552.3   | 1718.9  |
> | Orbis-FM | ST         |  360.9   |  246.0   |
> |            | DiT          |  345.4   |  274.2   |
> |            | CDiT         |  410.2   |  246.1   |
>
> *Table: FVD scores for small-scale versions of Orbis-MG (discrete) and Orbis-FM (continuous) with different architectures. The FVD is calculated over 60 generated frames on 200 samples of BDD validation set.*
>
> > **Q2** Additional metrics, such as inference speed, would be helpful for understanding the effectiveness of continuous tokens.
>
> As shown in the table below, we compare the inference speed of our discrete Orbis-MG model and the continuous Orbis-FM model. We also report the GPU (VRAM) memory requirements for both methods during inference. Orbis-MG shows better inference speed compared to the Orbis-FM model. However, since Orbis-FM achieves significantly better video generation performance, it remains the default choice despite the speed advantage of the discrete model.
>
> | Method     | FPS ↑ | VRAM (GB) ↓ |
> |------------|:-----:|:-----------:|
> | Orbis-MG   | 0.85  |     21      |
> | Orbis-FM   | 0.70  |     24      |
>
> *Table: Inference speed and VRAM requirements of the Orbis-MG and Orbis-FM models.*

---

> ### Comment · Reviewer_NJY5 · 2025-08-05
>
> Thanks for the rebuttal. It has addressed some of my concerns. However, a few issues remain — particularly that the core modeling approaches (e.g., autoregressive, flow-matching, MaskGIT) are primarily adaptations of existing techniques rather than novel contributions. Therefore, I will maintain my original score.

---

### Official Review · Reviewer_hx5U · 2025-07-03

**Clarity:** 3
**Significance:** 4
**Originality:** 3
**Rating:** 5
**Confidence:** 4

**Summary:**

- Video generative world models are critical for generation of training and evaluation data particularly for corner cases which are difficult to collect manually. This paper proposes a simple continous/discrete hybrid autoregressive (AR) model.
- The paper compares continous and discrete AR models and demonstrate that continous model works significantly better.
- The paper proposes that image quality metrics like FID is limited and adds a new trajectory based metric based on pose sequences generated by a recent CVPR'25 paper VGGT.
- The paper demonstrates significantly better results compared to state of art models in both FID and trajectory based metrics.
- The impressive impact of the paper is the simplicity and efficiency of the proposed approach with an order of magnitude smaller model and shorter trainings compared to Nvidia Cosmos model which is trained on 10,000 GPUs for three months.

**Questions:**

Please see the questions in weaknesses.

**Ethical Concerns:**

["NO or VERY MINOR ethics concerns only"]

**Final Justification:**

My comments were suitably addressed and will keep the original rating.

**Limitations:**

Limitations are adequately addressed in the discussion section.

**Paper Formatting Concerns:**

No formatting issues

**Quality:**

3

**Strengths And Weaknesses:**

Strengths:
- The paper explores a continous autoregressive model formulation based on Flow Matching. This is a new application of this idea for world models.
- The model is highly efficient in terms of training compared to the trend of using very large models for world model trainings. This will have a high impact to future work.
- Significantly better results are obtained compared to sota methods in terms of FID visual quality and trajectory quality metrics particularly for difficult scenarios like turns.

Weaknesses:
- It is not clear why the proposed method performs well for turns compared to other methods. Please add analysis or discussion to motivate why this happens.
- The dynamic agents like vehicles and pedestrians are quite important to simulate as the main goal of these videos is to train/test planning or end to end driving models. The performance of dynamic objects doesnt seem to be good relative to static background based on visual inspection of the qualitative videos in supplementary. It will be good to add discussion on this point.
- Vista paper already had a proposed trajectory metric, this is not discussed and compared to the new trajectory metric.

---

> ### Author Rebuttal · Authors · 2025-07-30
>
> Thank you for your positive feedback and thoughtful questions and comments.  We address each question and provide detailed discussions below.
>
> > **W1:** It is not clear why the proposed method performs well for turns compared to other methods. Please add analysis or discussion to motivate why this happens.
>
> - We speculated in the discussion section about two possible reasons for the failure of previous continuous driving models. We further investigated the hypothesis that parallel multi-frame prediction would impair long horizon generation, by training our smaller scale model to predict 12 frames at once (i.e. 2.4s ahead, like Vista and GEM). We found our model to work just as well with this setup, ruling out the hypothesis.
>
> - We could not verify the impact of pretraining biases due to involved heavy compute cost. However, we speculate that prior driving world models such as GEM and Vista, which initialize using general-purpose Stable Video Diffusion model, tend to inherit biases from pretraining data lacking scenarios like sharp turns that are critical for driving tasks. We also observe that the Cosmos model tends to produce smooth, drone-like trajectories, which may similarly stem from biases in the training data.
>
> - Additionally, we believe that it is not a single factor that boosts our model's performance. For instance, training with a flow-matching objective rather than a diffusion objective, such as the EDM used in Vista, can be advantageous. For example, previous works (e.g. *Diffusion Autoencoders are Scalable Image Tokenizers*, *Cascaded Diffusion Models for High Fidelity Image Generation'*) show that v-parameterization (used in flow matching) is more robust than $\epsilon$-parameterization of diffusion: the most related is "GameNGen" (Valevski et al., 2024), where one focus is long-horizon roll-outs in gaming environments. We also incorporated techniques such as adding noise to the context, GameNGen (Valevski et al., 2024) and dropping the context. We observed adding noise to the context improves FVD from 280 to 246 in a small-scale setting.
>
> - The exact reasons stay speculative, since direct comparisons that remove a single potential cause require training an entire new model, which costs 9k GPU hours.
>
> > **W2:** Suboptimal visual quality of dynamic objects relative to static background.
>
> -  We acknowledge that the current quality of the generated dynamics objects is suboptimal based on visual inspection.
>     However, we emphasize that the dynamic agents tend to appear in plausible places at a plausible time and with a realistic speed. This suggests that their internal representation seems to be better than their generated visual appearance suggests. We believe that seeing more of these agents in the data and concurrently scaling the model is necessary to achieve higher quality. Another possible direction could be training a larger diffusion decoder post-hoc (i.e., independent of the predictive model) to enhance visual fidelity. However, even such a decoder would need to see more of these dynamic agents during training to be effective.
>
> > **W3:** Vista paper already had a proposed trajectory metric, this is not discussed and compared to the new trajectory metric.
> -  The trajectory metric proposed in Vista measures the control effect of action conditioning. We will give proper credit to this. However, our proposed trajectory metric operates at a distribution-level, evaluating the quality and realism of ego-motion in the generated videos without action conditioning (Table 3 in paper).
>
> - We also evaluate the effect of conditioning on trajectories (Table 5 in paper) with the trajectory difference metric proposed by the authors of Vista, but we use a state-of-the-art off-the-shelf-model for inverse dynamics (VGGT, Wang et al. CVPR-25). This is consistent with the procedure used in GEM, where the SOTA DroidSLAM was used for inverse dynamics. We refer to both Vista and GEM while using this metric for evaluating action conditioning, in Line 296.

---

> > ### Comment · Reviewer_hx5U · 2025-08-09
> >
> > My comments were suitably addressed and I will keep my original rating of Accept.

---

### Official Review · Reviewer_DMVr · 2025-07-09

**Clarity:** 3
**Significance:** 2
**Originality:** 2
**Rating:** 4
**Confidence:** 3

**Summary:**

The paper addresses the difficulty of long-horizon video prediction in autonomous driving world models & proposes a new method called Orbis. There are 3 main contributions: a) It highlights the shortcomings of current driving world models on extended rollouts (e.g. turns) and introduces an evaluation benchmark with metrics to quantify long-horizon prediction quality. b) It proposes a hybrid discrete–continuous tokenization scheme that enables direct side-by-side comparison of discrete token prediction vs continuous prediction under the same token representation. Two corresponding world models are developed and trained from scratch on raw driving videos. c) Through this comparison, the authors find that the continuous autoregressive model significantly outperforms the discrete one in long-term fidelity, achieving SotA results on challenging driving scenarios (e.g. maintaining correct turning trajectories over 20-second rollouts). Notably, the proposed model uses only front-camera video and 469M parameters trained on 280 hours of data, yet surpasses prior larger models in both quality and rollout length. The paper also demonstrates a simple extension for ego-motion conditioning, allowing limited control over the generated trajectory.

**Questions:**

See the Weakness part above.

**Ethical Concerns:**

["NO or VERY MINOR ethics concerns only"]

**Final Justification:**

Thanks the authors for partially addressed my questions! After reviewing the rebutall, I decide to keep my original rating of Borderline Accept.

**Limitations:**

See the Weakness part above.

**Quality:**

3

**Strengths And Weaknesses:**

# Strengths

[Novel Approach and Insights]

1. The paper introduces a hybrid discrete–continuous tokenization strategy to fairly compare two paradigms of world models. This is a novel idea by using a unified token space for both a MaskGIT-based discrete model and a flow-matching continuous model, the study provides an apples-to-apples comparison of discrete vs. continuous generative approaches.

2. The authors identify a critical failure mode of existing driving models (generation for long horizons) and contribute an evaluation benchmark with corresponding metrics to capture this. They measure how closely generated trajectories can match the real trajectory over time.

[Comprehensive Experiments]

3. The experimental evaluation is extensive and thorough. The authors test their models on multiple datasets and scenarios, including a challenging turning-manuevers dataset (nuPlan-turns) and two other public driving benchmarks. Both short-horizon (6s) and long-horizon (20s) performance are reported, showing that the proposed continuous model outperforms all baselines. The paper also evaluates a fine-tuned version with steering inputs to demonstrate controllability, and it conducts ablation studies on the tokenizer design (e.g. testing the impact of adding DINO feature distillation or factorizing tokens).

4. The authors provide insightful analysis of their results, e.g. failure modes of the discrete model & comparisons with other continuous baselines (Vista, GEM, Cosmos, etc.). Overall, the experimental section not only demonstrates superior performance but also provides careful oblation studies that strengthen the conclusions.

[SotA Performance]

5. The proposed Orbis model achieves SotA results in driving video generation, while using fewer parameters and less training data. Orbis is particularly good in difficult scenarios like turning where prior methods often drift or collapse.

[High-Quality Writing]

6. The paper is well-written and easy to follow. The introduction clearly motivates the problem. The methodology has sufficient technical details and the experiments come with solid figures & tables. The authors also discussed limitations of the proposed methods.

# Weaknesses

[Limited Theoretical Insights]

1. The paper does not fully explain why diffusion-based video models struggle with long-horizon predictions. They proposed guesses on factors like pretrained biases or the parallel multi-frame prediction in those models, but no conclusive evidence is provided. This limits the theoretical insight of the work. The results show which approach (continuous AR) works better, but not fully why the other approach fails (aside from the discrete token “content copying” problems that they observed).

[Generation Fidelity Limitations]

2. Despite clear improvements, the fidelity and realism of the generated videos still have some limitations. The authors note that certain fine details are “not yet generated reliably”. For example, important objects like traffic lights and street signs often do not appear or are unclear in the output. Additionally, the behavior of other traffic participants in the generated videos is not always realistic. In practice, missing a red light or having a car break traffic rules in simulation could be problematic if the model were used for planning or training an agent.

[Insufficient Downstream Validation]

3. The scope of the paper is largely restricted to open-loop video prediction, and it does not demonstrate the model’s usefulness in downstream decision-making tasks. While long-horizon visual predictions are interesting on their own, it remains unclear how this world model would integrate into an autonomous driving system beyond simulation. The authors have not yet tested the model on downstream tasks, so we don’t know if the improved long-horizon generation actually leads to better planning or control performance.

[Benchmarking and Baseline Clarifications]

4. The comparison against the discrete methods might not capture the full potential of discrete token world models. The paper’s discrete baseline (Orbis-MG) is a novel implementation based on MaskGIT, since the best-performing discrete model (GAIA-1) is not open-sourced. While the authors did their best to build a strong discrete model, it’s worth noting that GAIA-1 (cited as having no issues with long drives) could not be included for direct comparisons. This leaves some uncertainty: the results show the continuous model is better than all public baselines, but we are not sure how it stands relative to the closed-source GAIA.

---

> ### Author Rebuttal · Authors · 2025-07-30
>
> Thank you for your positive feedback and thoughtful comments. We provide detailed discussion for all comments and answer each question below.
>
> > **W1:** [Limited Theoretical Insights] Further insights into potential failure factors for other models and future directions.
> -  We speculated in the discussion section about two possible reasons for the failure of previous continuous driving models. Meanwhile, we further investigated the hypothesis that parallel multi-frame prediction would impair long horizon generation, by training our smaller scale model to predict 12 frames at once (i.e. 2.4s ahead, like Vista and GEM). We found our model to work just as well with this setup, ruling out this hypothesis. We will add this result to the paper.
>
> - We could not verify yet the impact of pretraining biases and other architectural design choices due to a limited compute budget (every full training run costs 9k GPU hours). A few additional studies could be conducted in isolation in future work include: 1) comparing the v-parameterization of flow-matching with the $\epsilon$-parameterization used in of diffusion, as discussed in GameNGen paper [Valevski et al., 2024], and 2) investigating context augmentation strategies, such as injecting noise [Diffusion forcing Chen et al. 2024] or context dropout.
>
> > **W2:** [Generation Fidelity Limitations] Discussion on realism of the generated videos and model's planning ability.
>
> - Fully agreed that we are still some distance away from the point where the world model matches the quality needed for planning and training an agent in real driving scenarios. Our work is just another step in this direction.
> - We could only experiment at a resolution 288 x 512 due to a limited compute budget. Scaling up the resolution of this model input by $2\times$ should resolve some of the issues related to missing details. Also increasing the amount of training data would improve the model’s ability to capture more detailed state features. So far, our focus was on improving design choices before solving remaining issues by scaling the model, the resolution, and the data.
> - In this work, our focus is on two key challenges in learning driving world models - long-horizon video prediction and generalization to complex real-world scenarios, such as turning maneuvers and urban traffic. Long-horizon prediction serves as a proxy for evaluating how well the world model learns state transitions, while generalization to complex scenarios shows the model's ability to capture a diverse range of real-world dynamics.
>
> > **W3** [Insufficient Downstream Validation] Discussion on the scope of the paper.
>
> Most existing visual world models which can perform downstream decision-making tasks are limited to restricted or simplistic environments such as Atari, DMLab, or Minecraft. It is feasible to perform closed loop tasks in such settings either due to simpler requirements from the perception module or from the constrained environment setting. So far none of the current real video-based driving world models (including ours) are ready yet to be used in a closed-loop setting. In our work we focus on building a driving world model with robust state dynamics, which allows handling of difficult situations like turning maneuvers or urban driving. We observe that prior methods often struggle in such non-trivial yet common situations in an open-loop setting. We argue that developing models with stable and realistic transitions is critical before they can be effectively trained for planning and decision making. Our additional metrics beyond short-term FVD are early pointers, which we think are relevant for indicating the readiness, such as the degradation rate with increasing roll-out length and diverse plausible trajectories derived from the predicted video.
>
> > **W4:** [Benchmarking and Baseline Clarifications] Comparison with closed-source GAIA-1 model
> - GAIA-1 is not only closed-source; it is not even possible to run the model on new data. All evidence we have is from their paper and the few published videos. Thus, all discussion becomes quite speculative.
> - GAIA-1 having a large number of parameters (6.5B) may provide an advantage in enabling long rollouts. Additionally, they used a 2.6B-parameter decoder to reduce flickering.
> - Although it is not explicitly stated, the shift to a flow-matching objective in GAIA-2 may indicate the advantage of a continuous latent space over a discrete one for the GAIA model.

---

### Official Review · Reviewer_KDTT · 2025-07-11

**Clarity:** 3
**Significance:** 3
**Originality:** 2
**Rating:** 5
**Confidence:** 4

**Summary:**

The paper proposes a new video generative method for driving scenarios. The method allows to generate better (and more complex videos) considering a long horizon (> 5s). Thanks to diverse incremental improvements, the model yields state-of-the-art results despite having only 469M parameters and being trained on 280h of video. However, in my opinion, the results lack quality and resolution, making them impossible to reuse in other tasks.

**Questions:**

- Please add “Ours” in Fig.2. At a quick read, looks like Orbis is a different method.
- You mention that “high-resolution model operates at 512 ×288” for current standard that is not high-resolution. Could the model generate at HD or FHD. What are the limitations to generate real high-resolution content.
- Just to clarify, what are the FPS (frames per second) of the model? Assuming that you generate 20s clips, you are really generating 20 frames? Thus, 1 FPS?
- The high resolution model is trained for 10 epochs over 5 days on 72×A100 GPUs. Could you provide the loss function plots?
- Could you generate 1 minute clips? Just for the sake of understanding the limitations of the model and possible hallucinations.

**Ethical Concerns:**

["NO or VERY MINOR ethics concerns only"]

**Final Justification:**

I update my rating to accept. The authors tackled most of the important points of the rebuttal, although some points are open to discussion, the overall paper is solid and a good contribution to the recent trend of world models.

**Limitations:**

The authors discuss the limitations at the end of the paper. A general limitation is the misconception of “high-resolution”, claiming that 512 ×288 is high-resolution is not accurate.

The authors showcase how to generate videoclips of up to 20s, yet they do not explore longer horizons, even if the model fails the results are interesting for the community.

The authors evaluate the quality of the videos using FVD and FID, these metrics are often used by the generative community however, only attend at distributions. No real pixel-wise Video Quality Metrics (VQA) are used.

What is the purpose of the model? In the introduction you mention “ Recent driving world models [15, 1, 23] built on video diffusion models [4] have made major strides towards generating detailed content in high definition and at high frame-rate”
Let me doubt about the resolution and quality (high definition). From the real VQA quality perspective -pixel wise- the quality is very bad. I do not think these videos could be used for training other models, which is the main application of generating synthetic data.

**Paper Formatting Concerns:**

No Paper Formatting Concerns

**Quality:**

2

**Strengths And Weaknesses:**

Strengths
- The paper results are positive and the application is quite recent and relevant.
- I appreciate all the video results in the appendix
- The paper showcases good metrics in terms of FVD and FID (distribution metrics, not quality metrics)
- The authors show consistent 20s generation for autonomous driving videos.
- The method is more efficient than other methods in terms of data requirements.
- The authors discuss limitations and social impact.

Weaknesses
- No real pixel-wise Blind Video Quality Metrics (VQA) are used.
- The resolution and quality claims are overstatements. The maximum resolution is 512 ×288, and the quality is poor. The resultant videos could not be used in downstream tasks such as training a detector or segmentation model. What is the underlying purpose of such videos?
- The model is still heavy, trained over 5 days on 72×A100 GPUs. The cost of generating a video is not discussed, but probably ridiculously high.

---

> ### Author Rebuttal · Authors · 2025-07-30
>
> Thank you for the positive feedback and thoughtful questions. We address each question below and will incorporate the feedback in the revision.
>
> > **L1 & Q2:** Clarification on high-resolution terminology.
>
> - The term "high resolution" in the paper may be misleading, as it is used only to distinguish between the lower resolution (256×256) and the higher resolution (512×288) in our experiments. We will revise this terminology to avoid confusion.
> - The primary limitation for generating HD videos is the computational cost: training at HD resolution would be roughly six times more expensive due to having 6x more tokens, exceeding our current computational budget.
>
>
> > **L4 & W2:** Discussion on the purpose of the driving world model.
>
> We think generating realistic synthetic data can be a useful application, but it is generally not the primary objective of world models. As demonstrated in a number of influential works (e.g. *World Models* Ha et al. 2018, *Dreamer* Hafner et al. 2019, *Cosmos*, *VJEPA-2* Assran et al. 2025), the main purpose of world models is representation learning, planning, and policy learning. In this context, the quality of the state representation and the prediction of next states distribution is key. The decoded video is only one way to roughly assess to which degree this is successful, e.g. whether the model can successfully complete a turn without becoming unstable and without running into a steady state. The baselines we compare with (Vista, GEM, etc.) are conceived with planning as a possible application, however we observe that they often struggle in non-trivial yet mundane situations. In our work we prioritize building a world model with robust state representation and dynamics, which allows  handling such situations. One could train a more sophisticated high-resolution diffusion decoder post-hoc to improve visual quality, but this is not the scope of this paper.
>
>
> > **W3:** Training compute budget and inference costs for Orbis and other models.
>
> - Orbis training over 5 days on 72 A100 amounts to 9k A100 (40GB) GPU-hours. This is less than all baselines included in this work. For example, Vista was trained for 25k A100 (80GB) GPU-hours, DrivingWorld model for 18k A100-GPU-hours, and the general purpose Cosmos model for more than 20000k H100-GPU-hours.
> - The table compares the average times (fps) needed to generate a frame, and the required GPU memory. Orbis has the best throughput compared to all other methods. This advantage can be attributed to the smaller size of the Orbis model. By parallelizing Orbis' computation in batches we can achieve an higher throughput. GEM and Vista are based on the same architecture but use different sampling protocols trading off FPS and VRAM.
>
> | Method        | FPS ↑ | VRAM (GB) ↓ |
> |---------------|:--------:|:--------------:|
> | Cosmos        | 0.18   | 29           |
> | Vista         | 0.58   | 86           |
> | GEM           | 0.44   | 45           |
> | DW            | 0.25   | 10           |
> | Orbis (ours)  | 0.70   | 24           |
>
> *Table: Inference speed and VRAM requirements for different methods. All numbers are calculated on a H200 GPU.*
>
> > **W1 & L3:**  Pixel-wise video quality metrics.
>
> - We benchmarked Orbis using video quality metrics as requested. As shown in the tables below, we computed average PSNR and SSIM metrics over two shifted video windows of 10 frames (i.e. 2 seconds at 5fps) over 400 generated videos on the nuPlan-turns evaluation benchmark. We resized the generated videos of all methods to the same resolution for a fair comparison. Orbis performs marginally better than other methods over the first window. However, all methods converge to similarly low numbers in the second window.
>
> | Method        | PSNR (frames 0–9) | PSNR (frames 10–19) |
> |---------------|:----------:|:------------:|
> | Cosmos        |   17.29    |    13.38     |
> | GEM           |   14.85    |    13.73     |
> | Vista         |   15.04    |    13.70     |
> | DW            |   17.67    |    14.70     |
> | Orbis         |   18.72    |    14.75     |
>
> *Table: PSNR metrics computed over two shift windows (2 seconds each) from predicted frame 0 to 9 and frame 10 to 19.*
>
> | Method       | SSIM (frames 0–9) | SSIM (frames 10–19) |
> |--------------|:-----------------:|:-------------------:|
> | Cosmos       |       0.47        |        0.38         |
> | GEM          |       0.42        |        0.41         |
> | Vista        |       0.44        |        0.42         |
> | DW           |       0.44        |        0.38         |
> | Orbis        |       0.52        |        0.42         |
>
> *Table: SSIM metrics computed over two shift windows (2 seconds each) from predicted frame 0–9 and frame 10–19.*
>
> - We would like to remark that pixel-wise VQA metrics like PSNR and SSIM are not well-suited for evaluating the quality of generated videos. These metrics rely on exact correspondence between the real and the generated video, which is not sensible, as large parts of the future are non-deterministic and can take different plausible paths, especially on a pixel level.  This is reflected in the tables above, where all methods show similarly low numbers in the second window just after 2 seconds, both for PSNR and SSIM metrics.
>
> - Additionally, the visual fidelity of our model can surely be improved using a more powerful decoder or refiner network can be trained post-hoc, without having to change the world model itself -- similarly to what is done in other works (e.g. Cosmos, GAIA-1).
>
>
> > **Q5 & L2:** Observations on longer video rollouts.
>
> We are not allowed to provide longer clips for rebuttal due to the text-only format. In many cases, the model is able to produce longer drives, however, open-ended roll-outs expose two main failure modes: 1) the ego-vehicle stops at a red light, before an obstacle or a stationary vehicle, and does not restart. 2) the model hallucinates a scene change after too rapid shifts in context appearance. Restarting behavior becomes more common when training on more data. We observed this in previous scaling steps. The cause of the hallucinations is likely the limited context size. We agree that this is relevant for the community, and will add it to the discussion. We well also publish longer videos. Moreover, we will release the code and the model upon publication, so everybody can make all experiments they are interested in.
>
> > **Q4:** Orbis model loss function plot.
>
> We are unable to provide a loss function plot due to the text-only rebuttal format. The training loss curve follows the same trend as  Figure 13 in the DiT paper (Peebles et al. 2022). We can include the loss curve in the camera-ready version.
>
> > **Q1 & Q3:** Other clarifications.
>
> - Our model predicts at 5 fps. Therefore, 20s clips include 100 frames.
> -  In the camera-ready version, we will label our method as "Ours" in Fig. 2 to prevent any potential confusion.

---

> > ### Comment · Reviewer_KDTT · 2025-08-05
> > **Discussion with Authors**
> >
> > Thank you very much for your rebuttal. I am partially satisfied.
> > I understand that the purpose of world models (now) is basically experimental from a representations learning perspective, however, that has to be clear. The data produce by the models is still low-quality, and arguably unusable for any other downstream task.
> >
> > Note that not all the VQA metrics rely on pixel-wise correspondences, many blind video quality metrics evaluate aspects like the temporal consistency and level of details. These experiments should be easy to add and i suggest the authors to check these. Fo instance, you should check "Rich features for perceptual quality assessment of UGC videos", CVPR 2021 — I am not author or co-author of the paper, you can find many other metrics.
> >
> > The authors should include the aforementioned details in the final version of the manuscript. I suggest the authors to highlight the training benefits in their paper.

---

> > > ### Author Response · Authors · 2025-08-08
> > > **Additional VQA Results**
> > >
> > > Thank you for clarifying and suggesting the metric, which indeed evaluates videos based on inherent qualitative aspects, without ground truth frames. We looked into recent developments in the same line of work and found the DOVER score (Wu et al., _"Exploring Video Quality Assessment on User Generated Contents from Aesthetic and Technical Perspectives"_, ICCV2023), which is also used in the data curation/filtering pipeline of Cosmos.
> > > In the table below we report a comparison of the DOVER scores on the nuPlan Turns dataset, computed over 17s long videos. We include both the results for the full Cosmos pipeline (including the 7B-parameters diffusion refiner, "Cosmos+ref"), and for the pure Cosmos world model with a non-generative decoder.
> > > | Method | DOVER ↑ |
> > > |-----------------|----------|
> > > |Cosmos        |  19.94|
> > > |Cosmos+ref  |   28.99|
> > > |GEM             | 19.76|
> > > |Vista             | 21.14|
> > > |DW               |  21.92|
> > > |Orbis             |  21.34|
> > >
> > > The results show that all the models achieve similar scores, with the exception of Cosmos+ref, which outperforms the others thanks to its expensive refinement module. Interestingly, Cosmos alone performs slightly worse than most methods. This underlines the potential of boosting the visual fidelity of the generated videos post-hoc, if the application demands it - since the refinement step is independent from the training of the main world model.

---

### Comment · Area_Chair_WDf7 · 2025-08-03

Dear reviewers,

Now the rebuttal is available. Please discuss with authors and among reviewers asap.

Please try to come to a consensus on the key issues even though the rating can be different. Please feel free to let me know how I can help.

Best,

Your AC

---

### Decision · Program_Chairs · 2025-09-17

**Decision:**

Accept (poster)

**Comment:**

### Summary and Recommendation Justification

This paper presents an important advancement in driving world models, focusing on the critical and previously unsolved challenge of stable, long-horizon video prediction. The authors propose a continuous autoregressive world model based on flow matching, which demonstrates state-of-the-art performance in generating coherent and realistic driving scenarios for extended durations (20+ seconds), particularly in difficult situations like turning maneuvers. A key contribution of the work is a novel hybrid tokenizer that enables a rigorous, direct comparison between continuous and discrete generative approaches, ultimately providing strong evidence in favor of the continuous paradigm. The work is supported by extensive experiments, introduces new relevant evaluation metrics, and impressively achieves its results with a model that is significantly smaller and trained on less data than many of its competitors.

After a thorough and constructive discussion period, reviewers reached a clear consensus for acceptance. Reviewers consistently praised the work for its novel application of flow matching to world models, its impressive empirical results, and its high efficiency compared to the state of the art. The paper makes a well-supported, and timely contribution to a very active area of research.

### Detailed Justification

**Strengths:**

1.  **State-of-the-Art Long-Horizon Generation:** The paper's primary empirical contribution is its demonstration of robust, long-horizon video generation in complex driving scenarios. This directly addresses a well-known failure mode of previous world models, which often degrade or collapse over shorter timeframes. As noted by multiple reviewers, the ability to maintain coherence and realistic ego-motion over 20+ seconds is a substantial step forward for the field.

2.  **Novel and Rigorous Comparative Methodology:** A standout contribution is the design of a hybrid tokenizer that facilitates a fair comparison between continuous (flow-matching) and discrete (MaskGIT-based) generative models within the same token space. This is a valuable scientific contribution that provides clear evidence for the advantange of the continuous approach in this domain, a finding that is of high interest to the generative modeling community.

3.  **Efficiency and Simplicity:** In a field increasingly dominated by massive models trained on vast computational resources, this work is a compelling demonstration of efficiency. As highlighted by Reviewer hx5U, the proposed model achieves comparable results with an order of magnitude less compute and a much smaller parameter count than leading industrial models like Cosmos. This makes the approach more accessible and provides a strong signal about the importance of architectural and objective function choices over sheer scale.

### Addressing Initial Concerns

The initial reviews raised valid concerns primarily related to the evaluation of video quality, the precise purpose of the generated videos, and the lack of certain experimental details. The authors' rebuttal was effective in addressing these points:

*   **Video Quality and Purpose:** In response to Reviewer KDTT's concerns, the authors clarified that the primary purpose of a world model is representation learning and planning, not necessarily pixel-perfect synthetic data generation. Critically, they also conducted **new experiments** during the rebuttal, adding several VQA metrics (PSNR, SSIM, and the blind DOVER score) to their evaluation, which provided a more nuanced picture of video quality and showed their model to be competitive.
*   **Computational Cost and Inference Speed:** The authors provided a detailed breakdown of training and inference costs, demonstrating that their model is not only more data-efficient but also faster at inference time compared to all baselines.
*   **Architectural Ablations:** To address concerns from Reviewer NJY5 about the fairness of the discrete vs. continuous comparison, the authors ran new ablations across three different Transformer architectures, showing that the continuous model's superiority was consistent and not an artifact of a single architectural choice.

The authors' willingness and ability to conduct significant new experiments during the rebuttal period substantially strengthened the paper. The reviewers were largely satisfied with the responses. While some open questions remain, such as a deeper theoretical explanation for the failure of other models and the potential for downstream planning applications, these are seen as avenues for future work rather than critical flaws in the current contribution.

In short, this paper presents a well-executed study with clear and impactful results. It advances the state of the art in a challenging domain, offers a novel and valuable methodological comparison, and does so with a remarkably efficient model.